# Exploring the Mechanisms of Recovery in Anorexia Nervosa through a Translational Approach: From Original Ecological Measurements in Human to Brain Tissue Analyses in Mice

**DOI:** 10.3390/nu13082786

**Published:** 2021-08-13

**Authors:** Philibert Duriez, Ida A. K. Nilsson, Ophelia Le Thuc, David Alexandre, Nicolas Chartrel, Carole Rovere, Christophe Chauveau, Philip Gorwood, Virginie Tolle, Odile Viltart

**Affiliations:** 1Institute of Psychiatry and Neuroscience of Paris (IPNP), University of Paris, INSERM UMR-S 1266, F-75014 Paris, France; p.duriez@ghu-paris.fr (P.D.); p.gorwood@ghu-paris.fr (P.G.); virginie.tolle@inserm.fr (V.T.); 2GHU Paris Psychiatry and Neurosciences, Hospital Sainte-Anne, F-75014 Paris, France; 3Department of Molecular Medicine & Surgery, Karolinska Institutet, Centre for Eating Disorders Innovation (CEDI), Medical University, Karolinska Institutet, S-17176 Stockholm, Sweden; ida.nilsson@ki.se; 4CNRS UMR 7275, Institute of Molecular and Cellular Pharmacology (IPMC), University of Nice-Sophia Antipolis, F-06560 Valbonne, France; ophelia.lethuc@helmholtz-muenchen.de (O.L.T.); rovere@ipmc.cnrs.fr (C.R.); 5INSERM U1239, Laboratory of Neuronal and Neuroendocrine Differentiation and Communication, University of Normandie, UNIROUEN, F-76821 Mont-Saint-Aignan, France; david.alexandre@univ-rouen.fr (D.A.); nicolas.chartrel@univ-rouen.fr (N.C.); 6Marrow Adiposity and Bone Laboratory (MABLab), University of Littoral Côté d’Opale, CHRU Lille, F-62327 Boulogne sur Mer, France; christophe.chauveau@univ-littoral.fr; 7Faculty of Sciences and Technologies, University of Lille, F-59650 Villeneuve d’Ascq, France

**Keywords:** physical activity, heart rate, anorexia nervosa, body mass index, hypothalamus, neuropeptides, ghrelin, leptin, animal model

## Abstract

Anorexia nervosa (AN) is a severe eating disorder where caloric restriction, excessive physical activity and metabolic alterations lead to life-threatening situations. Despite weight restoration after treatment, a significant part of patients experience relapses. In this translational study, we combined clinical and preclinical approaches. We describe preliminary data about the effect of weight gain on the symptomatology of patients suffering from acute AN (*n* = 225) and partially recovered (*n* = 41). We measured more precisely physical activity with continuous cardiac monitoring in a sub-group (*n* = 68). Using a mouse model, we investigated whether a long-term food restriction followed by nutritional recovery associated or not with physical activity may differentially impact peripheral and central homeostatic regulation. We assessed the plasma concentration of acyl ghrelin, desacyl ghrelin and leptin and the mRNA expression of hypothalamic neuropeptides and their receptors. Our data show an effect of undernutrition history on the level of physical activity in AN. The preclinical model supports an important role of physical activity in the recovery process and points out the leptin system as one factor that can drive a reliable restoration of metabolic variables through the hypothalamic regulation of neuropeptides involved in feeding behavior.

## 1. Introduction

Anorexia nervosa (AN) is a complex psychiatric disorder emerging most commonly in adolescence and young adulthood whose etiology has remained poorly understood up to now, even if metabolic and endocrine aspects are now considered as key pathophysiological determinants [1,2,3,4,5]. The severe weight loss due to food restriction is usually combined with excessive physical activity [6,7,8]. The identification of factors influencing the onset, duration and quality of the recovery process is a research priority for eating disorders [9]. One out of two patients with AN relapses within a year following inpatient treatment and approximately 20% of patients experience either recurrent patterns of remission and relapses or have a chronic outcome [10,11,12]. Weight gain results in an initial improvement in AN symptomatology and constitutes a major first step in the medical treatment of AN. Weight gain stabilization, within a range of global symptomatology improvement, is a therapeutic priority, which has to be maintained for months. Partial recovery is probably a frequent evolution after an acute relapse, with persistent qualitative food restriction, problematic physical exercise and/or body image alteration [13].

Previous studies aimed at determining biological markers of the remission processes. Leptin and ghrelin are key candidates well known for their role in regulating feeding behavior, energy metabolism and motivational and rewarding processes associated with feeding and running [4,5]. Leptin is an adipokine, mainly expressed in adipocytes [14], for which plasma concentrations increase in proportion to body fat stores [15]. This hormone is involved in weight regulation and various biological functions such as immune function, reproduction, bone formation, anxiety and physical activity [5,16,17,18]. Ghrelin is a pleiotropic hormone, mainly released by the X/A-like endocrine cells in the oxyntic glands of the gastric fundus and initially described as a growth hormone secretagogue [19,20]. Similar to leptin, ghrelin is known to participate both in metabolic regulations such as food initiation, glucose homeostasis or gastric emptying and in complex behaviors such as anxiety, the reward process or compulsive physical activity [21,22,23,24,25]. Two ghrelin isoforms have been described: acylated ghrelin (AG), which has an n-octanoylated serine in position three and desacyl ghrelin (DAG, [26,27]). Both forms have distinct biological functions [21,28].

At the clinical level, basal plasma leptin concentrations have recently been considered as a reliable marker of stable recovery with a better predictivity than body mass index [29]. Furthermore, in a pilot study of three patients with restrictive AN, a two-week treatment with metreleptin (recombinant human leptin) generated positive outcomes such as reduction in physical activity, of weight phobia and of negative emotional feelings [30]. In patients with AN, plasma variations of AG and DAG during the weight restoration period might be considered as reliable biomarkers of a successful recovery [31]. Indeed, we recently showed that DAG plasma concentration after weight restoration is associated with instable recovery and an increase in physical activity after discharge [32]. However, an accurate measurement of physical activity in AN remains difficult and data are still lacking, limiting translational benefits from preclinical models such as the activity-based rodent model of AN [33].

Both ghrelin and leptin target brain structures involved in the homeostatic control of food intake. More specifically, the hormones receptors are expressed on neurones located in the hypothalamus, namely the orexigenic neurones co-expressing agouti-related peptide (AgRP) and neuropeptide Y (NPY) and the anorexigenic neurones expressing proopiomelanocortin (POMC). These neurones constitute critical nodes of a circuit that sense key metabolic cues to regulate energy metabolism and feeding behavior [34,35,36,37,38]. Recently, both AgRP and POMC neurones were shown to also regulate circuitries controlling non-feeding behavior, such as the reward pathway, anxiety, stereotypic behaviors, compulsive exercise and delay discounting all related to AN [34,39,40,41,42,43,44,45]. These neurones are precisely located in the ventromedial part of the hypothalamic arcuate nucleus (ARC) and are considered as primary targets of convergent peripheral hormones [46,47]. Recently, Florent et al. (2019), using magnetic resonance imaging, clearly demonstrated alterations in the hypothalamic response to a meal in patients with AN compared to control and subjects with constitutional thinness [48].

The current imaging techniques are not precise enough to capture the changes occurring at subcellular levels. For this purpose, using preclinical models is a prerequisite to better understand mechanisms involved in the remission or relapse of AN. Several animal models are now available to assess the various aspects of AN [32,49,50,51,52]. Using the “activity based-anorexia” (ABA) model, Verhagen et al. (2011) demonstrate that central leptin injections in the lateral ventricle or local injections of leptin into the ventral tegmental area suppress the running wheel activity [53]. In particular, using a mouse model that recapitulates metabolic signatures of AN, we recently showed that plasma DAG concentrations were positively correlated with diurnal physical activity during a two-week nutritional recovery [32]. These data let us hypothesize that chronic food restriction, associated with a moderate physical activity, might differentially impact the expression of hypothalamic neuropeptides targeted by ghrelin and leptin.

In this translational study, we combined a clinical and preclinical approach to find potential explanations to remission or relapse in partially recovered patients with AN. We first describe initial original data about physical activity in both acutely ill and partially recovered patients with AN in order to highlight the candidate mechanisms of the recovery process. Secondly, using the FRW (food restricted with wheel) model, we investigate whether short (15 days) or long-term (65 days) food restriction associated or not with physical activity, may differentially impact the mRNA expression of hypothalamic neuropeptides and their receptors. As we will also measure the plasma concentration of AG, DAG and leptin, we will present the hypothalamic mRNA expression of the ghrelin receptor and leptin receptor.

## 2. Materials and Methods

### 2.1. Experiment 1: Clinical Assessment

All female patients with a current diagnosis of AN (*n* = 266) were enrolled in the present study after referring to a specialized center for eating disorders, the CMME (Clinique des Maladies Mentales et de l’Encéphale, Paris, France). All patients signed a written consent for their participation in research protocol, according to the Helsinki Declaration. The diagnosis of AN was established by a clinical interview conducted by a trained clinician using a validated instrument (MINI-S), following DSM-5 diagnostic criteria [54]. According to DSM-5, patients who previously met all criteria for AN but for a sustained period were not low in body weight were diagnosed as partial recovery AN (prAN) [54].

Clinical and biological data were obtained after a two-day multidisciplinary assessment of the patient. We gathered age, current BMI, adult minimal BMI and illness duration. Minimal BMI reflects the severity of weight loss and undernutrition state during the course of AN according to DSM-5 [54]. We also collected classical nutrition markers: albumin (normal range: 35*–*52 g/L) and ferritin (normal range: 15*–*150 µg/L) blood levels.

Eating disorder symptoms were assessed using the Eating Disorder Inventory-2 [55]. The EDI2 consists of 91 items measuring 11 psychopathological dimensions. High scores represent greater levels of psychopathology. Depression and anxiety scores were measured with the self-report Hospital and Anxiety Depressive Scale (HADS) [56]. A high score (max 21) indicates a high level of anxiety or depression, respectively. Compulsivity and obsessionality were monitored with the Yale Brown Scale [57]. A high score (max 20) shows a high level of obsessionality and/or compulsivity.

We recently developed the evaluation of cardiac activity through a 72 h monitoring, using electrocardiogram Holter monitoring (ePatch*^®^*). On the 68 patients evaluated, 55 were patients with acute AN (aAN) and 13 were patients with prAN. The Holter monitor was fitted on day 1 and removed three days later. Maximum heart rate (HRmax) was calculated for each patient with the formula: 220-age. We calculated proportion of time when the heart rate was in four ranges defined as: bradycardia (below 60 beats/min), physiological (between 60 beat/min and 50% of HRmax), moderate activity (between 50% and 60 % of HRmax), high activity (higher than 60% of HRmax).

### 2.2. Experiment 2: Preclinical Approach

#### 2.2.1. Animals

Adult C57BL/6J female mice (7 wk old, average initial body weight 18.8 ± 0.15 g; Charles River Laboratories, L’Arbresle, France) were housed two per cage in all the procedures. Mice are social animals; thus, in an ethological perspective and for the welfare of the animals, we decided to maintain our mice two per cage to avoid stress isolation and hypothermia induced by our long-term caloric restriction protocol. They were kept in a pathogen-free barrier facility maintained at 21.5 °C with a 12:12 h dark–light cycle (lights on at 07:30 a.m.). During one week of acclimation, mice were weighed every day to become used to handling and had free access to water and standard chow diet (4.30% fat, 22.30% protein and 51.20% carbohydrate; Special Diet Service RM3; Dietex, Essex, UK). Mice were randomized into three groups: The experimental group “food restriction and wheel” (group FRW), mice were placed in a cage equipped with a free running wheel (INTELLIBIO, Seichamps, France) and exposed first to a 30% quantitative food restriction for three days then followed by 50% quantitative food restriction for all the duration of the protocol. This restriction was calculated every day from the total food consumed by each mouse in the group of mice fed ad libitum (group AL) the previous day, by weighing the whole pellets in the feeder. Food (one pellet per mouse) was distributed directly into the cage every day at 6:30 p.m. This restriction was also applied to another experimental group: A pair-fed group or group FR (food-restricted). Body weights were monitored. Control of body weight was conducted to ensure that two mice in the same cage had similar feeding behavior and activity. If evident differences were noted between the two mice, the data obtained from this cage were excluded from analysis. The daily locomotor activity of FRW mice was assessed in their home cage equipped with a wheel (diameter: 230 mm; width: 50 mm; 1 revolution = 0.72 m) and linked to a computer system (ActiviWheel Software; INTELLIBIO, Seichamps, France). No mice died in any protocol. All experiments were carried out in accordance with the European Communities Council Directive (86/609/ EEC) and approved by the Regional Ethics Committee of Nord-Pas de Calais of Lille, France (protocol no. CEEA 392012).

#### 2.2.2. Protocols

Three protocols were performed. At the end of each protocol, the mice were sacrificed. The short-term protocol (ST) consisted of food restriction for two weeks (FR, *n* = 6; FRW, *n* = 6; AL, *n* = 6; 3 cages per group) while in the long-term protocol (LT), the food restriction was maintained for ten weeks (FR, *n* =8; FRW, *n* = 8; AL, *n* = 8; 4 cages per group). Finally, in another set of animals, a two-week nutritional recovery phase was performed, following the long-term protocol (LT-Rec; FR, *n* = 6; FRW, *n* = 6; AL, *n* = 6; 3 cages per group). For this purpose, FR and FRW had ad libitum access to standard diet. Free access to the running wheel was maintained for FRW mice. Body weight was measured every two days in all protocols while food intake was measured daily. Body weight gain was calculated to evaluate the percentage of loss between D0 and D65 (100-((BW at D65/BW at D0) × 100)) and the percentage of gain after two weeks of recovery (100-((BW at D86/BW at D65) × 100)). The locomotor activity was assessed daily from the running wheel linked to a computer system that measured interval counts (10 min) per mean wheel revolution (ActiviWheel Software; INTELLIBIO, Seichamps, France). Data were extracted with an excel macro (Microsoft Office Standard, 2016) to obtain cumulative activity or day/night activity.

#### 2.2.3. Euthanasia and Tissue Collection

At the end of the protocols, mice were sacrificed in the morning between 8:00 a.m. and 11:00 a.m. They were deeply anesthetized with an overdose of ketamine (100 mg/kg) and xylazine mix (20 mg/kg). Blood was collected through cardiac puncture with a 1 mL syringe and kept at 4 °C for ~2 h in neutral tubes containing p-hydroxy-mercuribenzoic acid (PHMB 0.4 mM final), a serine protease inhibitor, until centrifugation (8000 rpm for 10 min, 4 °C, Centrifuge 5414 R; Eppendorf, Hamburg, Germany). Samples were rapidly centrifuged (1000× *g* for 10 min, 4 °C), to gain plasma aliquots. An aliquot of around 100 μL was immediately acidified with HCl (0.1 N final) to preserve ghrelin acylation; the residual plasma was kept for other assays. Plasma aliquots were then frozen in dry ice before being stored at −80 C until they were assayed. From brains, the hypothalamus was carefully dissected, immediately frozen in liquid nitrogen and stored at −80 °C until analysis. Subcutaneous and perigonadic adipose tissues were gently dissected and weighed.

#### 2.2.4. Immunocytochemistry

In another set of experiments, mice were deeply anesthetized with an overdose of ketamine (100 mg/kg) and xylazine mix (20 mg/kg) after a short-term food restriction protocol (FR, *n* = 2; FRW, *n* = 2; AL, *n* = 2) and transcardially perfused with 100 mL of 0.9% saline followed by 100 mL of 4% paraformaldehyde in 0.1 mol/L phosphate-buffered solution (PBS, pH 7.4). Brains were removed, postfixed at 4 °C overnight in 4% paraformaldehyde in PBS and frozen in dry ice before processing. Immunocytochemical procedure was previously described in Bouret et al., 2004. Coronal hypothalamic sections (20 μm) were obtained with a Cryostat (Leica) and collected in PBS. Sections were incubated for 48 hours in a rabbit anti-AgRP (polyclonal rabbit antiserum against AgRP, 1:4000; Phoenix Pharmaceutical, Belmont, CA). The primary antibodies were localized with a biotinylated goat antirabbit IgG (1:600, 90 min, Vector Laboratories). Tyramide signal amplification (TSA) was accomplished by placing the sections in an avidin–biotin solution (Vectastain) for 1 h, followed by incubation in TSA solution for 20 min. according to the manufacturer’s instruction (TSA-Indirect kit, New England Nuclear Life Science). Deposited biotin was detected with Alexa 488-conjugated streptavidin (Molecular Probes). Sections were mounted and coverslipped with buffered glycerol (pH 8.5). Quantification of the fiber density was performed in the paraventricular nucleus (PVN, main location of ARC AgRP projections) using a Leica SP-confocal microscope equipped with a 10× objective (numerical aperture, 0.40; working distance, 360 μm). The density of immunolabeling was evaluated with Image J (NIH, 1987, https://imagej.nih.gov/ij/download.html (accessed on 12 August 2021) for 3 sections per nucleus per animal. The procedure was carried out on each image plane in the stack, and the values for all image planes in a stack summed. The resulting value was an estimate of the AgRP fiber density in the PVN.

#### 2.2.5. Blood Assays

All samples were analyzed in duplicates. Plasma leptin was measured using an ELISA kit (R&D Systems Quantikine Europe, Abingdon, UK). Intra- and interassay coefficients of variations were 4.4 and <4.7%, respectively. Plasma acyl-ghrelin and des-acyl ghrelin concentrations were evaluated by specific EIA (A05118 for the acylated form and A05117 for the des-acyl form; Bertin Bioreagent, Montigny le Bretonneux, France). Intra- and inter-assay coefficients of variations were 6.1% and 5.7% for AG and 5.5% and 4.8% for DAG, respectively.

#### 2.2.6. RT-PCR Extraction and Analysis

Total hypothalamus RNAs were extracted in guanidinium thiocyanate and phenol-chloroform according to Chomczynski and Sacchi method [58]. Two micrograms of RNA were treated with DNase I (Roche Diagnostics, Penzberg, Germany) then reverse-transcribed using SuperScript III (Invitrogen, Thermo Scientific, Waltham, MA, USA) and real-time PCR was performed using SYBR Green mix (2X, Roche Diagnostics, Meylan, France). The glyceraldehyde 3-phosphate dehydrogenase (GAPDH) was used as internal control and housekeeping gene to normalize gene expression. The purity of the PCR products was assessed by dissociation curves. The amount of target cDNA was calculated by the comparative threshold (Ct) method and expressed by means of the 2^−^^ΔΔ^^Ct^ method. PCR primers were described in Table 1.

### 2.3. Statistical Analysis

The clinical data were expressed as means ± SD and analyzed with jamovi (the jamovi project 2021, version 1.6.23). Analysis of normality of variances was tested by Shapiro–Wilk test. Mann–Whitney test was used to compare aAN and prAN populations. Spearman’s rank correlation coefficient was used to analyze correlations between clinical dimensions (Shapiro–Wilk *p* < 0.05). For the cardiovascular data, raw data were analyzed through a homemade algorithm using Python. The preclinical data were expressed as means ± SEM. GraphPad Prism 5.01 (Abacus Concepts, Berkeley, CA, USA) was used to analyze the data and generate graphs. Analysis of normality was tested by Shapiro–Wilk test. Equality of variance was tested by Levene’s test. Statistical analysis was performed using one-way ANOVA or two-way ANOVA when appropriate, followed by a Tukey post hoc test when the *p* value of the ANOVA was significant (*p* < 0.05). When equality of variance was violated, Games–Howell post hoc test was used. We eliminated three individual values related to mRNA expression of GHSR (group AL—short term; group FR—long term) and of leptin receptor (group FRW-LT recovery) using the z-score test which deciphers whether value is aberrant or not (Z-score = (value − mean) / SD)).

## 3. Results

### 3.1. Experiment 1. Preliminary Data: Clinical Investigation of Partial Recovery

#### 3.1.1. Clinical Characteristics: Acute AN versus Partially Recovered AN Patients

We recruited 266 patients suffering from acute (aAN, *n* = 225) or partial recovery (prAN), *n* = 41) AN. Illness duration was similar between aAN and prAN (U = 3274, *p* = 0.199), as was the level of anxiety (U = 2969, *p* = 0.069), depression (U = 3485, *p* = 0.663), obsession (U = 3061, *p* = 0.117) and compulsion (U = 3410, *p* = 0.525). On the other hand, patients with aAN had a lower minimal BMI (U = 1956, *p* < 0.001), a lower total EDI-2 score (U = 3133, *p* = 0.007) especially with the EDI-2 sub-scores, drive for thinness (U = 3030, *p* = 0.004), bulimia (U = 3290, *p* = 0.017), body dissatisfaction (U = 2750, *p* < 0.001), perfectionism (U = 3407, *p* = 0.044), interoceptive awareness (U = 3143, *p* = 0.008) and emotional dysregulation (U = 3185, *p* = 0.011). Lastly, albumin (U = 1670, *p* < 0.001) and ferritin (U = 845, *p* = <0.001) were higher in patients with prAN (Table 2).

#### 3.1.2. Continuous Cardiac Monitoring, Past and Actual BMI and Illness Duration

Among the 266 patients, 68 had a 72 h continuous heart rate monitoring: 55 aAN and 13 prAN. This measure was used to assess the level of bradycardia and of physical activity. Comparison between the sample with continuous cardiac monitoring and the total sample did not show significant differences except for anxiety in the prAN group (10.83 ± 4.24 with monitoring versus 13.87 ± 3.45, *p* = 0.019) and illness duration in the AN group (10.45 ± 7.36 with monitoring versus 9.09 ± 8.76; *p* = 0.032) and aAN group (10.54 ± 7.62 with monitoring versus 8.88 ± 8.79; *p* = 0.034, Table 3).

We found a negative correlation between the lowest BMI and high intensity physical activity (rho = −0.271, *p* = 0.029). Illness duration was negatively correlated to bradycardia (rho = −0.298, *p* = 0.015) and positively correlated to moderate physical activity (rho = 0.277, *p* = 0.024). All correlations for the AN group are reported in Table 4.

We then analyzed the differences between patients who have reached the normal BMI range for a sustained period (prAN, *n* = 13) and acute patients (aAN, *n* = 55). Compared to the prAN group, the aAN group had more time with bradycardia (31.5% versus 13.8%; U = 200, *p* = 0.02), less time with a physiological heart rate (51.2% versus 65.4%; U = 183, *p* = 0.009) and an equivalent time at moderate (10.1% versus 11.1%; U = 261, *p* = 0.181) or high (U = 262, *p* = 0.186) physical activity (Figure 1).

In aAN, lifetime minimal BMI, but not current BMI, was negatively correlated to a high level (rho = −0.332, *p* = 0.016) and moderate level (rho = −0.279, *p* = 0.045) physical activity. Illness duration was negatively correlated to bradycardia (rho = −0.423, *p* = 0.002) and positively correlated to moderate (rho = 0.324, *p* = 0.018) and high (rho = 0.307, *p* = 0.025) intensity of physical activity (Table 5). In the prAN group, only a trend for a negative correlation was found between minimal BMI and high intensity of physical activity (rho = −0.525, *p* = 0.065, Table 6).

### 3.2. Experiment 2. Preclinical Investigation

#### 3.2.1. Body Weight, Food Intake, Physical Activity and Fat Mass Evolution

At D0, body weight was not significantly different between all groups (*n* = 6 per group): AL: 18.22 g ± 0.29; FR: 18.22 g ± 0.24; FRW: 18.03 g ± 0.12, F(2,15) = 0.21, *p* = 0.815. The day before nutritional recovery, at day 65, the body weight of FR and FRW mice was significantly decreased compared to AL mice, respectively (14.07 g ± 0.14 and 14.65 g ± 0.09 versus 21.42 g ± 0.37; F(2,15) = 290.2, *p* < 0.001; Figure 2A). At the end of the food restriction protocol (D65), the body weight gain calculated from D0 was −22.8% and −18.9% for FR and FRW mice, respectively, with a significant difference (F(2,15) = 280.9, *p* < 0.001) between AL and FR mice (*p* < 0.001, Figure 2B).

The nutritional recovery led to a rapid gain in body weight reaching +55.9% and +48.8% for FR and FRW mice, respectively, at the end of the protocol (D86, Figure 2C) with a significant difference (F(2,15) = 165, *p* < 0.001) between AL and FR mice (*p* < 0.001, Figure 2C). However, despite a full recovery in body weight within 3 days of ad libitum feeding (D68), we noticed that FRW mice showed a transient body weight reduction between D68 and D71 which was significantly different compared to FR mice with an interaction time × group (F(12,90) = 67,19, *p* < 0.001, Figure 2A). Indeed, AL mice showed a significantly higher body weight than FR mice at D65, D66, D67 (*p* < 0.005), and FR mice gained significantly more weight than FRW mice at D69, D70, D71 (*p* < 0.05). Such difference can be attributed to the quantity of food ingested during all the nutritional recovery period (Figure 2D). Two-way ANOVA indicated a protocol effect (F(5,10)= 7.9, *p* = 0.003) and a group effect (F(2,10)= 4.7, *p* = 0.04) with a significant difference at D66 (*p* = 0.0004), D67 (*p* = 0.03), D68 (*p* = 0.003), D69 (*p* = 0.002) and D70 (*p* = 0.01) between AL and FR mice. Surprisingly, the binge-eating behavior in the FR and FRW groups followed different patterns, with a longer duration for the FR group at D69 (*p* = 0.01) and D70 (*p* = 0.005).

FRW mice exhibited 24 h running wheel activity which decreased along the long-term protocol (Figure 2E, F(9,52) = 5.57, *p* < 0.0001). However, the percentage of time spent in activity during the day increased progressively in the long-term protocol (Figure 2F). Along the two weeks of nutritional recovery, the FRW mice showed an increased activity at D69 compared to the first day of recovery (D66, *p* < 0.05) which could explain the transient decreases in food intake and body weight (Figure 2E). However, nutritional recovery was accompanied by a decrease in the percentage spent in diurnal running wheel activity to a complete cessation of this activity at D81 (Figure 2F).

Finally, in parallel with the changes in feeding behavior and physical activity, FR and FRW mice showed a differential recovery of their fat mass (Figure 3A,B). Indeed, for both subcutaneous (SCAT, Figure 3A) and perigonadic (PG, Figure 3B) fat mass weight, two-way ANOVA indicated a group effect (SCAT, F(2,60) = 23.75, *p* < 0.0001; PG, F(2,61) = 36.88, *p* < 0.0001), a protocol duration effect (SCAT, F(2,60) = 8.94, *p* = 0.0004; PG, F(2,61) = 47.07, *p* < 0.0001), a trend for interaction protocols x groups (F(4,60) = 2.46, *p* = 0.054) for the SCAT and an interaction protocols x groups (F(4,61) = 3.53, *p* = 0.012) for PG. In the short-term protocol, and FR mice had a significant decrease in SCAT (*p* = 0.0004) and PG (*p* < 0.001) compared to AL mice. This decrease was higher in FRW mice than FR mice only for PG (trend, *p* = 0.09). The long-term protocol exacerbated these changes in fat mass (AL versus FR: SCAT, *p* = 0.0003; PG, *p* = 0.0002) and a trend only for SCAT (FR versus FRW, *p* = 0.054). Finally, only FR mice fully recovered their PG fat mass after two weeks of refeeding (*p* = 0.004).

#### 3.2.2. Ghrelin and Leptin Assays

Plasma ghrelin and leptin concentrations were evaluated in the three protocols as nutritional sensors (Table 7). Two-way ANOVA analyses exhibited a protocol effect for AG (F(2,43) = 23.78, *p* < 0.001) and DAG (F(2,43) = 6.41, *p* = 0.004) but not for leptin (F(2, 51) = 0.99; *p* = 0.3785), a group effect for AG (F(2,43) = 4.85, *p* = 0.013), DAG (F(2,43) = 6.95; *p* = 0.002) and leptin (F(2,51) = 22,28, *p* < 0.0001) and an interaction between protocols and groups for AG (F(4,43) = 5.19, *p* = 0.002) DAG (F(4,43) = 2.41, *p* = 0.064) but not for leptin (F(4, 51) = 1.55 ; *p* = 0.203).

In the short-term protocol, AG and DAG plasma concentrations were significantly higher in FR mice compared to AL mice (AG, *p* = 0.002; DAG, *p* = 0.013), but no significant difference seen between FR and FRW mice (AG, *p* = 0.133; DAG, *p* = 0.772). Finally, leptin plasma concentrations were significantly lower for FR mice (*p* = 0.029) compared to AL mice, but no difference was seen between FR and FRW mice (*p* = 0.811). In the long-term protocol, AG and DAG plasma concentrations were not significantly different (AG, AL versus FR: *p* = 0.936, FR versus FRW: *p* = 0.225; DAG, AL versus FR: *p* = 0.130, FR versus FRW, *p* = 0.910). However, leptin plasma concentrations were lower for FR mice compared to AL mice (*p* = 0.006) and similar for FR mice compared to FRW mice (*p* = 0.912). After nutritional recovery, mice exhibited no differences of AG and DAG plasma concentration (AG, AL versus FR: *p* = 0.902, FR versus FRW: *p* = 0.774; DAG, AL versus FR: *p* = 0.577, FR versus FRW, *p* = 0.449) whereas leptin plasma concentration remained lower for FR mice compared to AL mice (*p* = 0.029) and similar for FR mice compared to FRW mice (*p* = 0.106).

In a longitudinal perspective, AL mice displayed the same profile for AG, DAG and leptin plasma concentration in the three protocols (AG, short-term versus long-term: p = 0.451; long-term versus LT-rec: *p* = 0.959; short-term versus LT-rec: *p* = 0.399. DAG, short-term versus long-term: *p* = 0.062, long-term versus LT-rec; *p* = 0.146, short-term versus LT-rec: *p* = 0.415. leptin, short-term versus long-term: *p* = 0.398; long-term versus LT-rec: *p* = 0.995; short-term versus LT-rec: *p* = 0.398). For FR mice, AG plasma concentrations were significantly higher in short-term compared to long-term (*p* < 0.001) and LT-Rec (*p* < 0.001) protocols and similar between long-term and LT-rec (*p* = 0.948). DAG plasma concentrations were significantly higher in both short-term (*p* = 0.021) and long-term (*p* = 0.032) protocols compared to the LT-Rec protocol and similar between short-term and long-term (*p* = 0.802). The absence of a protocol effect on leptin plasma concentrations did not allow a post hoc analysis. For FRW mice, AG plasma concentrations were higher in the short-term than in the long-term protocol (*p* = 0.008) and LT-rec protocol (*p* = 0.003) with a trend of to be lower between long-term and LT-Rec protocols (*p* = 0.057). DAG plasma concentrations exhibited no differences between short-term and long-term protocol (*p* = 0.711) but were lower in LT-rec than in the short-term protocol (*p* = 0.011) and in the long-term protocol (*p* = 0.002). Leptin plasma concentrations were significantly lower in the LT-rec protocol compared to long-term protocol (*p* = 0.012) but exhibited no differences for short-term vs. long-term (*p* = 0.072) and LT-rec (*p* = 0.685).

#### 3.2.3. Immunocytochemical Detection of AgRP

Considering the metabolic changes induced by the short-term protocol [32,49], we further investigated whether a chronic food restriction might induce noticeable changes in the immunocytochemical AgRP labeling distribution in ARC and PVN of the hypothalamus. Surprisingly, in the ARC, the FRW mice displayed AgRP-immunoreactive cell bodies not seen in AL and FR mice (Figure 4), whereas the AgRP fiber density was not significantly different in the PVN (data not shown, *p* = 0.6).

#### 3.2.4. mRNA Expression of AgRP/NPY and Melanocortin Systems in the Hypothalamus

The changes in the immunolabeling of AgRP in FRW mice, associated with the variations in the plasma concentrations of ghrelin and leptin as well as a distinguished fat mass recovery compared to FR mice led us to investigate whether these changes might lead to a differential regulation in the expression of neuropeptides regulated by these hormones and involved energy metabolism (Figure 5).

Two-way ANOVA analysis indicated a time effect for AgRP and POMC mRNA expression (Figure 5A,C; AgRP, F(2,46) = 8.86, *p* = 0.0006; POMC, F(2,45) = 8.45, *p* = 0.0008), a group effect for AgRP, NPY and POMC mRNA expression (Figure 5A–C; AgRP, F(2,46) = 5.83, *p* = 0.005; NPY, F(2,47) = 6.51, *p* = 0.003 POMC, F(2,45) = 13.54, *p* < 0.0001) and an interaction protocol x group for AgRP and POMC and a tendency for NPY (AgRP, F(4,46) = 4.46, *p* = 0.003; NPY, F(4,47) = 2.29, *p* = 0.074, POMC, F(4,45) = 4.41, *p* = 0.004).

In the short-term protocol, AgRP and NPY mRNA expressions were significantly higher in FR mice compared to AL mice (*p* < 0.05) and in FRW compared to FR mice (trend for AgRP, *p* = 0.09). POMC mRNA expression was significantly decreased only for FRW mice (*p* = 0.007) compared to FR mice. The long-term protocol exacerbated significantly the changes in AgRP and NPY mRNA levels especially for the FR mice compared to AL mice (AgRP, *p* = 0.003; NPY, *p* = 0.002) and FRW mice (AgRP, *p* = 0.05; NPY, *p* = 0.04). Indeed, compared to AL mice, the long-term protocol led to a 10-fold and a 3-fold higher AgRP and NPY mRNA levels in FR mice, whereas the short-term protocol led to a 3-fold and a 2-fold higher expression, respectively. Finally, the POMC mRNA levels were significantly lower, especially for FR mice compared to AL mice (*p* < 0.001). Nutritionally recovered mice presented a normalization of NPY and POMC mRNA levels. For AgRP mRNA levels, one-way ANOVA revealed a tendency with FRW mice expressing more AgRP than FR mice (*p* = 0.07).

When we considered the evolution of mRNA expression of each group in the three protocols, we noticed first that AL mice displayed the same profile for each period and for the three investigated neuropeptides (Figure 5). Secondly, along the three protocols, FR and FRW mice did not display similar changes in the neuropeptide mRNA expressions. For AgRP, significant differences were both noted for FR and FRW groups: mRNA AgRP levels of FR mice were significantly increased in the long-term protocol compared to the short-term (tendency, *p* = 0.08) and LT-Rec (*p* = 0.017) protocols, whereas for FRW mice, it was significantly lower in the LT-Rec protocol than in the short-term (*p* = 0.004) and long-term protocols (*p* = 0.003). For NPY, only FRW mice showed significant changes with a higher mRNA expression in the short-term than in the long-term (*p* = 0.017) and LT-Rec (*p* = 0.009) protocols. Finally, for POMC, once again, the FR and FRW mice did not show a similar evolution of their mRNA expression: FR mice displayed a significant decrease in their POMC mRNA expression between the short-term and long-term protocols (*p* = 0.04) and a significant increase between the long-term and the LT-Rec (*p* = 0.02) protocols, whereas FRW mice showed a POMC mRNA level significantly higher in the LT-Rec protocol than for the short-term (*p* = 0.001) and long-term protocols (*p* = 0.03).

AgRP/NPY and POMC neurons were targeted by ghrelin and leptin. Therefore, we aimed at evaluating whether the mRNA expression of these receptors was modified in the three protocols (Figure 6). A two-way ANOVA analysis indicated a time effect for GHSR, Y1-R and MC4-R mRNA expression (GHSR, F(2,37) = 8.67, *p* = 0.0008; Y1-R, F(2,40) = 10.21, *p* = 0.0003; MC4-R, F(2,42) = 3.1, *p* = 0.055), a group effect for GHSR, Lept-R, Y1-R, Y5-R and MC3-R mRNA expression (GHSR, F(2,37) = 6.87, *p* = 0.003; Lept-R, F(2,40) = 10.89, *p* = 0.0002, Y1-R, F(2,40) = 7.73, *p* = 0.001; Y5-R, F(2,41) = 3.46, *p* = 0.04; MC3-R, F(2,42) = 12.41, *p* < 0.0001) and an interaction protocol x group for GHSR, Lept-R, Y1-R, Y5-R, MC3-R and MC4-R mRNA expression (GHSR, F(4,37) = 3.78, *p* = 0.01; Lept-R, F(4,40) = 3.89, *p* = 0.009, Y1-R, F(4,40) = 8.62, *p* < 0.0001; Y5-R, F(4,41) = 4.06, *p* = 0.007; MC3-R, F(4,42) = 5.21, *p* = 0.002; MC4-R, F(4,42) = 6.07, *p* = 0.0006).

In the short-term protocol, GHSR, Lept-R, Y1-R and Y5-R mRNA expressions were significantly lower in FR mice compared to AL mice (*p* < 0.05), and no significant difference was noted between FR and FRW mice. In contrast, MC3-R mRNA expression was significantly higher only for FR mice compared to AL (*p* = 0.024) and FRW mice (*p* = 0.002) with no significant changes for MC4-R mRNA expression. As previously mentioned, the long-term protocol induced no major changes compared to short-term: GHRS, AL versus FR (*p* = 0.03); Lept-R, AL versus FR (*p* = 0.005); Y1-R, AL versus FR (*p* < 0.0001); MC3-R, AL versus FR (*p* = 0.03) and FR versus FRW (*p* = 0.008); whereas the mRNA levels of Y5-R and MC4-R of FR mice were significantly higher compared to FRW mice (Y5-R, *p* = 0.016) and to AL (MC4-R, *p* = 0.015) and FRW mice (MC4-R, *p* = 0.003). The nutritional recovery led to a return to normal values for mRNA expression of GHSR. However, FR mice showed a differential recovery compared to FRW mice for the Lept-R, Y1-R and Y5-R mRNA levels which were higher than, respectively, in FRW mice (Lept-R, *p* = 0.006; Y1-R, *p* = 0.016; Y5-R, *p* = 0.006) and in AL mice (Y1-R, *p* = 0.021). Finally, the MC3-R and MC4-R mRNA expressions were significantly higher in FR mice compared to AL mice (MC3-R, *p* = 0.02; MC4-R, *p* = 0.009).

In AL mice, except for GHSR and Y1-R mRNA expression whose levels were significantly higher in the short-term (*p* < 0.05) than in long-term protocol, we noticed a similar profile of expression for each time and for the four other investigated receptors. In the three protocols, FR and FRW mice did not display similar changes in the receptor mRNA expressions. For GHS-R, significant differences were only observed for FRW mice with a significant decrease in the mRNA expression in the long-term protocol compared to short-term (*p* = 0.004). For Lept-R, the mRNA levels were significantly higher both for FR and FRW compared to the LT-Rec protocol between short-term (FR, *p* = 0.0006; FRW, *p* = 0.002) and long-term (FR, *p* < 0.001; FRW, *p* = 0.003). Y1-R, FR and FRW did not show the same evolutions: the mRNA levels in FR mice were higher in the LT-Rec protocol between the short-term (FR, *p* = 0.003) and long-term (FR, *p* = 0.0004) protocols, whereas for FRW, the Y1-R mRNA levels in LT-Rec returned to values obtained in the short-term protocol (ST versus LT, *p*= 0.002; LT versus LT-Rec, *p* < 0.001). Finally, no significant changes were noted for FR mice in the three protocols concerning the mRNA expression of Y5-R, MC3-R and MC4-R. It was not the same for FRW. Indeed, the Y5-R mRNA levels in the LT-Rec were similar to values obtained in the short-term protocol (ST versus LT, *p*= 0.0001; LT versus LT-Rec, *p* = 0.002), whereas the mRNA levels of MC3-R and MC4-R were higher between the short-term (MC3-R, *p* < 0.0001; MC4-R, *p* = 0.0005), long-term (MC3-R, *p* < 0.0001; MC4-R, *p* = 0.0001) and LT-Rec protocols.

## 4. Discussion

Anorexia nervosa is characterized by a high percentage of relapse after weight recovery. To prevent worsening and disastrous outcomes, we need to understand the mechanisms sustaining AN symptomatology and to decipher the factors which could contribute to restore an appropriate homeostatic balance between physical activity food intake, and body weight.

In the present study, we combined a clinical evaluation of acutely ill and partially recovered patients with restrictive-type AN using a new ecological cardiac monitoring device to assess physical activity with preclinical mouse model mimicking several aspects of AN in which we could record running wheel activity as an index of physical activity. For the first time, we showed an effect of a history of undernutrition, uncovered by the minimal BMI, on the level of physical activity in AN. Furthermore, the preclinical model supports an important role of physical activity in the recovery process, and indicates the leptinergic system as one factor that can drive a reliable restoration of metabolic variables through the hypothalamic regulation of neuropeptides involved in feeding behavior.

### 4.1. Physical Activity in Acute AN and prAN

To examine indirectly the effect of weight gain on AN symptomatology, we compared, in a treatment-seeking sample, patients with aAN and prAN. We found comparable anxiety and depressive symptoms and a more severe global symptomatology severity assessed with EDI-2 in the prAN group. Some dimensions appeared to be more severe, especially those connected to body image and bulimia. It is, thus, possible that the prevalence of binge/purging type of AN is higher in the prAN group (not controlled for), as indicated by a higher minimal BMI in this group. It is also possible that this particular group may constitute the most severe patients with prAN, since others may be less susceptible to seek treatment in a specialized center. It is known that partial recovery is a common evolution in AN [13].

To specifically address physical activity, we presented here original data of continuous cardiac monitoring permitting an ecological assessment of physical activity through the remission process. Most of the knowledge about physical activity in AN was found through declarative data or accelerometry, which may underestimate physical activity. Original and objective data on this core part of symptom of AN are still lacking [33]. The objective assessment of physical activity in daily life remained difficult until the recent development of new devices. Tools covering just a part of the physical activity (accelerometry) or the patient him- or herself (auto- and hetero- questionnaires) may underestimate physical activity. In contrast, cardiac monitoring presents the advantage of capturing all physical activities, including movement-free activity. These new approaches will hopeful allow a translational benefit with preclinical models to better understand, assess and treat excessive physical activity in AN (Figure 7). Our initial results described a link between both a history of low body weight and a high duration of illness, and a high level of physical activity. Variations in BMI are usually considered as a major event of in the recovery process and are illustrated in daily practice by weight history. We showed here that a history of severe weight loss, and not current BMI, is associated with higher levels of current physical activity. It could be considered such as a sequelae of the undernutrition state. Weight restoration is a cornerstone of treatment [59] and a strong prognostic factor for treatment efficacy [60]. However, persistence of symptoms after weight gain, illustrated here through treatment-seeking patients with prAN, has to be further investigated. The interaction between physical activity, caloric restriction and weight loss has been widely described and explored with various rodent models mimicking several physiological, metabolic and endocrine aspects of AN [32,50,51]. We further investigated such models here.

### 4.2. Impact of Moderate Physical Activity during Chronic Food Restriction on Recovery

After a long period of chronic food restriction, FR and FRW mice recovered in three days, a body weight similar to AL mice by adopting a binge-eating-like behavior. However, the running wheel activity performed by FRW mice induced several changes compared to FR mice such as a failure to recover perigonadic fat mass and a transient decrease in food intake associated with a slight increase in physical activity and a transient reduction in body weight. The moderate physical activity adopted during the long-term protocol might be adaptive considering the low body weight of these food restricted mice and may have a protective function to avoid rapid weight gain during nutritional recovery. Indeed, using a rat model of AN (ABA model), Giles et al. (2016) demonstrated that a rapid regain in body weight was metabolically unhealthy as it resulted in an increased lipid accumulation in the liver such as we can observe in the refeeding syndrome described in patients with AN [61].

These differences between FR and FRW might be attributed to plasma leptin concentrations, which were lower only in FRW mice at the end of the nutritional recovery compared to the long-term protocol. These decreased leptin levels were possibly the consequence of perigonadic fat mass that remained low after recovery in FRW mice, despite a comparable body weight with FR mice. Similar data were obtained with another mouse model mimicking anorexia, the “separation-induced weight loss” model (SWL). However, SWL mice, which did not have access to a running wheel such as FRW mice, recovered a fat mass similar to AL mice [50]. Furthermore, a recent analysis of the evolution of plasma leptin levels in patients with AN at inclusion, at discharge and two months later, showed that low leptin levels at discharge were associated with poor outcome [29]. For more than a decade, rodent and human studies have pointed to a causal link between excessive activity and low circulating leptin levels [17,18,62,63] leading to a recent preliminary case series using a recombinant analog of human leptin to treat with success three seriously ill patients with AN [30]. Thus, leptin might be considered as a potential reliable indicator of recovery linked to a physical activity assessment. The evolution of plasma concentrations of ghrelin was not different between FR and FRW mice in our conditions. As a metabolic sensor, AG and DAG were more sensible to the nutritional state than to the presence of physical activity [4,28].

### 4.3. Nutritional Recovery Differentially Impacts Hypothalamic Activity

When we considered the primary hypothalamic targets of ghrelin and leptin, namely, the AgRP/NPY and POMC neurons, we also uncovered noticeable changes. FRW mice displayed after two weeks of food restriction a distinguishable phenotype from FR mice. They displayed an accumulation of AgRP in neuronal cell bodies as also described in the genetic model of AN, the anx/anx mice [64,65], but without any changes in the fibers density at the level of PVN, one of the main target of AgRP arcuate neurons. This could be related to the higher level of AgRP mRNA in the hypothalamus after the short-term protocol described here, but also in another study using rats submitted to a short-term food restriction associated with running wheels (ABA model; [66]). A recent elegant study pointed out that selective arcuate NPY/AgRP activation, using DREADD technology, significantly increased not only the food intake, but also acute moderate-intensity exercise [37]. Furthermore, considering MC3/MC4 receptors which are naturally antagonized by AgRP [67], their mRNA expression was strongly diminished during the food restriction period, only in FRW mice. These data may reflect the impact of physical activity rather than the nutritional state. The NPY mRNA expression followed the evolution of AgRP, with an adapted and stable increase for FRW mice, whereas in FR mice this augmentation was enhanced during the duration of the protocol. The orexigenic effect of NPY is relayed through Y1-R and Y5-R in the PVN and lateral hypothalamic area [68,69]. These receptors are also abundantly present on POMC neurons [70], where Y1-R stimulation leads to an inhibition of electrical activity [71] and Y5R activation inhibits α-MSH release [72]. Their mRNA expressions were decreased except for Y5-R, for which expression increased only in FR mice during the long-term food restriction, suggesting once again a differential impact of physical activity on the regulation mechanisms of feeding. The down-regulation of Y1-R expression might be considered as a consequence of the upregulation of NPY and the associated feedback control. The mRNA expression of POMC mirrored the relative gene expression of orexigenic peptides, with a decrease which appeared earlier for FRW mice than for FR mice. This adapted decrease was also described in the ABA model [66].

Finally, the chronic food restriction was also accompanied by a reduced mRNA expression of GHSR and leptin receptor. To our knowledge, our study is the first to measure hypothalamic GHSR expression in an AN-like rodent model. Despite an initial increase in its expression in the FRW mice after two weeks of protocol, probably due to a feedback control induced by high circulating AG levels, GHSR mRNA expression decreased when the protocol was prolonged. We can hypothesize that a continuous release of AG and DAG might establish a desensitization of the hypothalamic GHSR. Concerning the leptin receptor, a similar decrease occurred in the relative gene expression of the leptin receptor in the SWL model during a long-term food restriction protocol [73]. This decrease might be paralleled with the total breakdown of fat masses.

The two-week of ad libitum refeeding led to a recovery of the mRNA expression of AgRP, NPY and POMC. However, this was not the case for the receptors. The upregulation of MC3/MC4-R mRNA expression in both FR and FRW mice may have reflected the changes in energy status since these two groups of mice gained more than 50% of their body weight during the duration of the nutritional recovery period. The mRNA expression Y1- and Y5-R of FRW mice returned to AL basal values, which was not the case for FR mice. Y1 and Y5 receptors are known to mediate the hyperphagic effect of NPY [74]. Indeed, the simultaneous ablation of both receptors in the germline Y1Y5 receptor double knockout (Y1Y5−/−) mice led to reductions in spontaneous and/or fasting-induced food intake. This indicates that food intake in mice requires the coordinated action of both the Y1 and the Y5 receptors [75]. Thus, here, we hypothesized that this increase observed in FR compared to FRW mice might have been a consequence of the hyperphagic behavior observed after refeeding, especially for FR mice. In parallel, the leptin receptor mRNA expression remained low only in FRW mice, whereas GHSR mRNA expression was stabilized. Once again, food restriction associated with moderate physical activity, as it was the case for FRW mice, induced a differential recovery compared to sole food restriction. The impact of physical activity was particularly salient on the leptin system and on related pathways. To link our preclinical data with the clinical reality, further studies are now needed to investigate, for example, the behavior of these mice faced to a food challenge, such as high caloric food.

### 4.4. Limitations

Our study of partial remission patients was based on treatment-seeking patients only. This sampling bias has to be considered as a limitation to address the effect of renutrition on AN symptom. These individuals may be a subgroup of partial recovery patients. The included patients may be among the most severe patients with partial recovery, as they seek help even though having normal weight. A longitudinal design could bring more accurate data to reinforce the effects observed in the present study.

Nevertheless, our sample remains informative. We presented for the first-time initial data obtained from continuous cardiac monitoring. To better distinguish the activity profiles of our AN patient cohorts, we need to develop precise algorithms.

Concerning the mouse model, the main limitation was the difficulty to gather all the data obtained in each protocol, because the brain analyses of peptides and receptors need to use three different groups of mice. There is currently no consensus about the nutritional recovery in mouse model mimicking the metabolic aspects of AN. To our knowledge, the analysis on peripheral and central aspects of nutritional recovery after a long-term protocol of food restriction associated or not with physical activity is lacking. Finally, we deliberately did not include another control group consisting of mice with ad libitum food access and a running wheel (ALW). Indeed, our previous works have underlined that AL and ALW mice displayed similar behavior in anxiety tests and comparable changes in endocrine and metabolic parameters evolution during the protocols [32,49,52].

## 5. Conclusions

Taken together, our translational approach allowed us to study the role of physical activity in the modulation of feeding behavior in the context of nutritional recovery in AN. The present clinical data bring a new ecological measure of cardiac activity that reflects the physical profile of patients with AN. In combination with the undernutrition history and likely biomarkers, it will allow us to better characterize and treat aAN and prAN patients to reach a full recovery (Figure 7).

The preclinical data point out several potential central mechanisms, which can be key elements in the recovery processes. Further studies using molecular tools are now necessary to better apprehend the role of the leptin system in chronic food restriction and recovery. Our data reflecting the central role of leptin and related pathways in nutritional recovery are supported by recent papers on AN, since systemic leptin levels after weight restoration predict the short-term [29] and long-term outcomes of patients with AN [76]. The results obtained here allow us, considering a potential combination of biomarkers, to take into account in order to validate a definition of remission: the physical activity profile according to an undernutrition history, biological assays and functional metabolic neuroimaging. Indeed, on hypothalamic biosensors of nutritional status in mice might be paralleled with the structural and functional changes observed in the hypothalamus of patients with AN using proton magnetic resonance spectroscopy and diffusion tensor imaging [48]. The neurobiological and neuroendocrine data shed new light on the mechanisms of nutritional recovery and open new perspectives to prevent relapse in AN.

## Figures and Tables

**Figure 1 nutrients-13-02786-f001:**
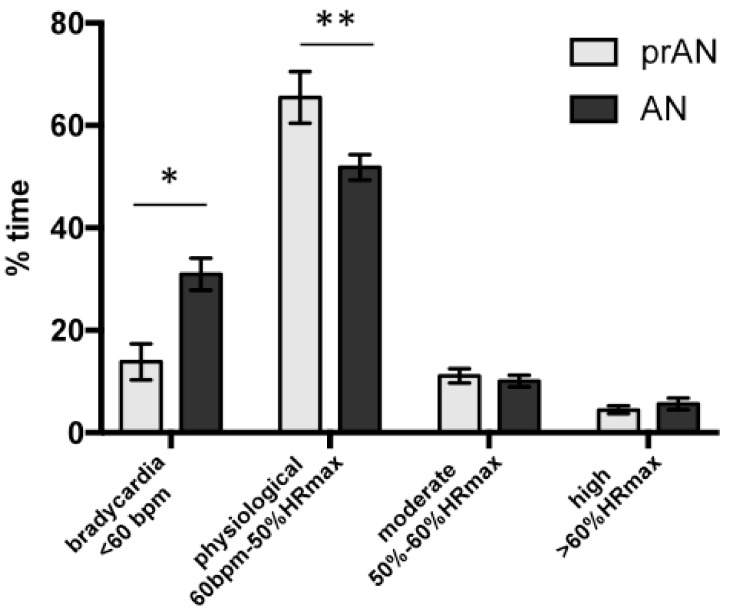
Data obtained from 72 h cardiovascular recording allowing to determine the percentage of time spent in bradycardia, physiological, moderate and high physical activity in patients with acute anorexia nervosa (aAN, *n* = 55) and partially recovered AN (prAN, *n* = 13). bpm: beat per minute; HRmax: maximum heart rate (220-age). * *p* < 0.05; ** *p* < 0.01.

**Figure 2 nutrients-13-02786-f002:**
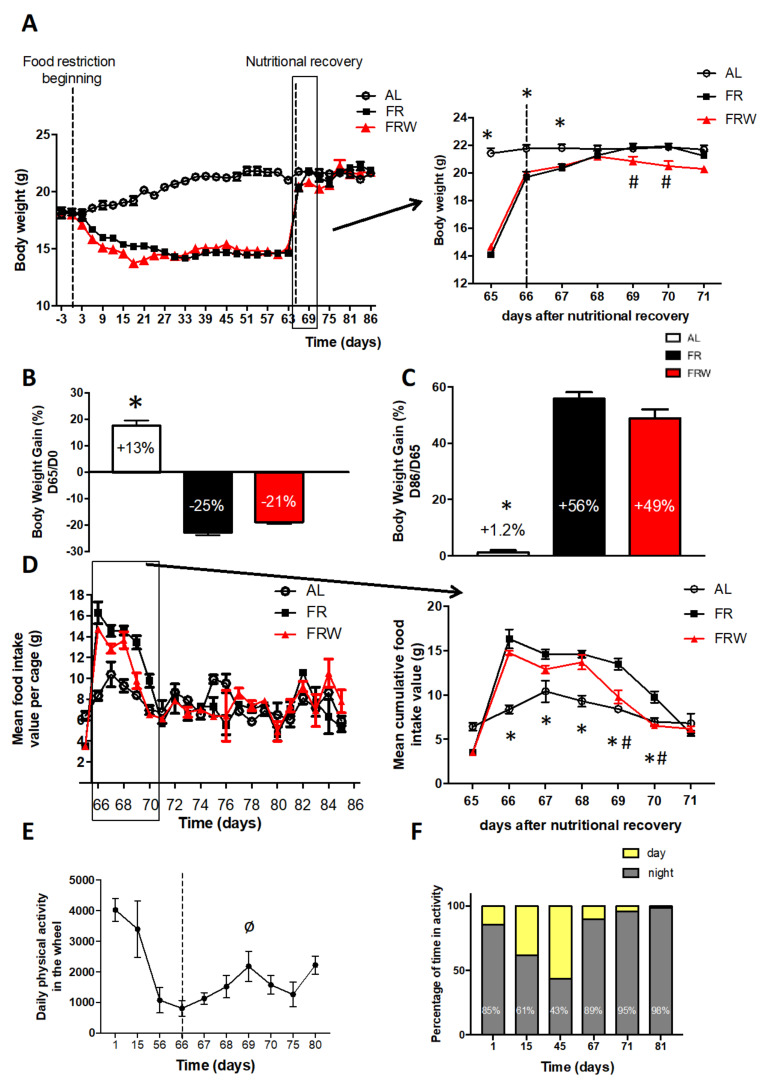
Body weight evolution (**A**–**C**), food intake during nutritional recovery (**D**) and daily physical activity (**E**,**F**) in the three protocols: short-term, long-term, long-term and nutritional recovery. AL: *ad libitum*; FR: food restriction; FRW: food restriction and wheel. Data represent mean ± SEM (*n* = 6–8 per group); * *p* < 0.05 AL vs. FR; # *p* < 0.05 FR vs. FRW. ø, *p* < 0.05 Day 66 vs. Day 69.

**Figure 3 nutrients-13-02786-f003:**
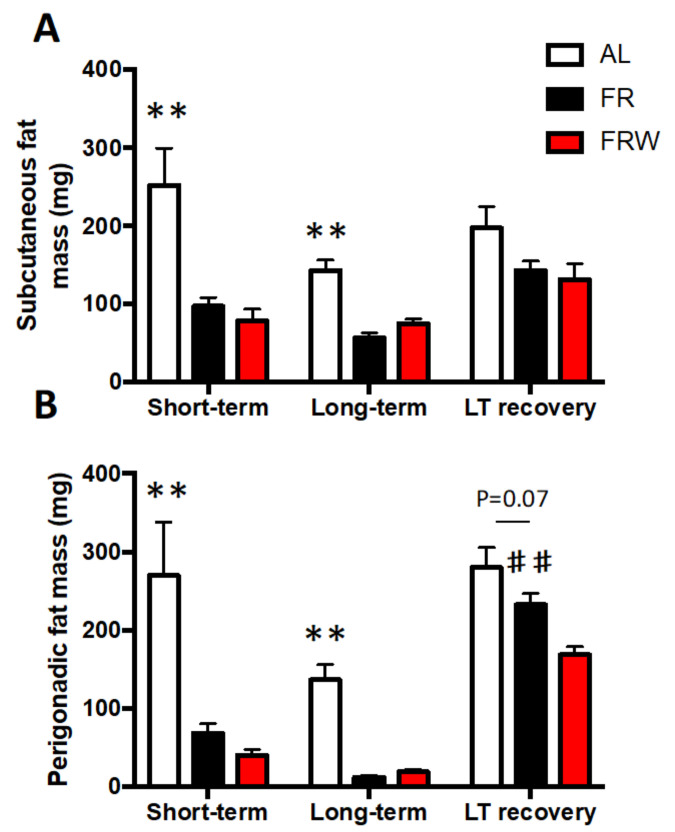
Adipose tissue weight. (**A**) Subcutaneous fat mass and (**B**) perigonadic fat mass weighed in the three protocols: short-term, long-term, long-term (LT) and nutritional recovery. AL: *ad libitum*; FR: food restriction; FRW: food restriction and wheel. Data represent mean ± SEM (*n* = 6–8 per group); ** *p* < 0.01 AL vs. FR; ## *p* < 0.01 FR vs. FRW.

**Figure 4 nutrients-13-02786-f004:**
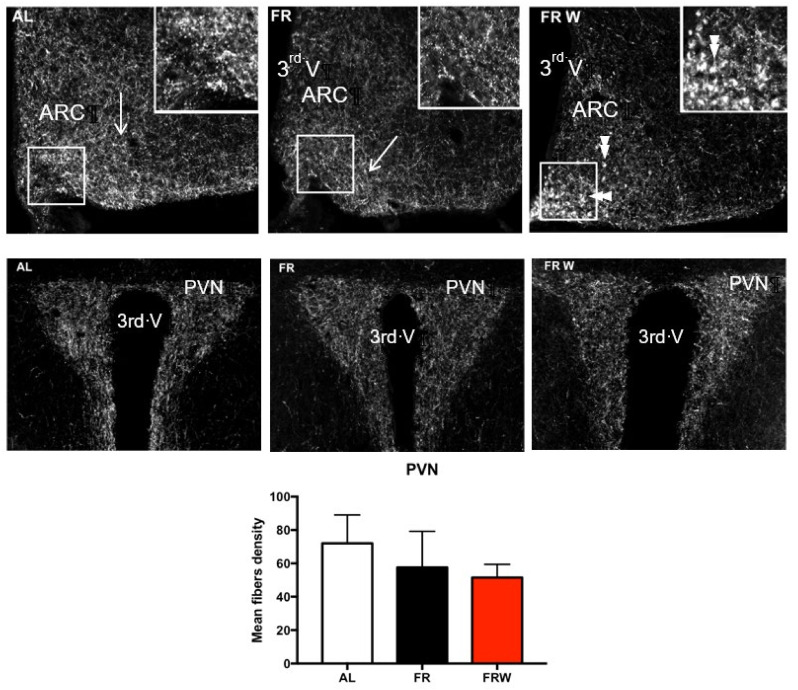
Immunohistochemical detection of AgRP (agouti-related peptide) in hypothalamic arcuate nucleus (ARC), fibers (arrows) and cell bodies (double arrows) and paraventricular nucleus (PVN) in AL, FR and FRW groups submitted to a short-term protocol. The three groups did not show any significant changes in the AgRP fibers density in the PVN. AL: *ad libitum*; FR: food restricted; FRW: food restricted with wheel; 3rd V: 3rd ventricle.

**Figure 5 nutrients-13-02786-f005:**
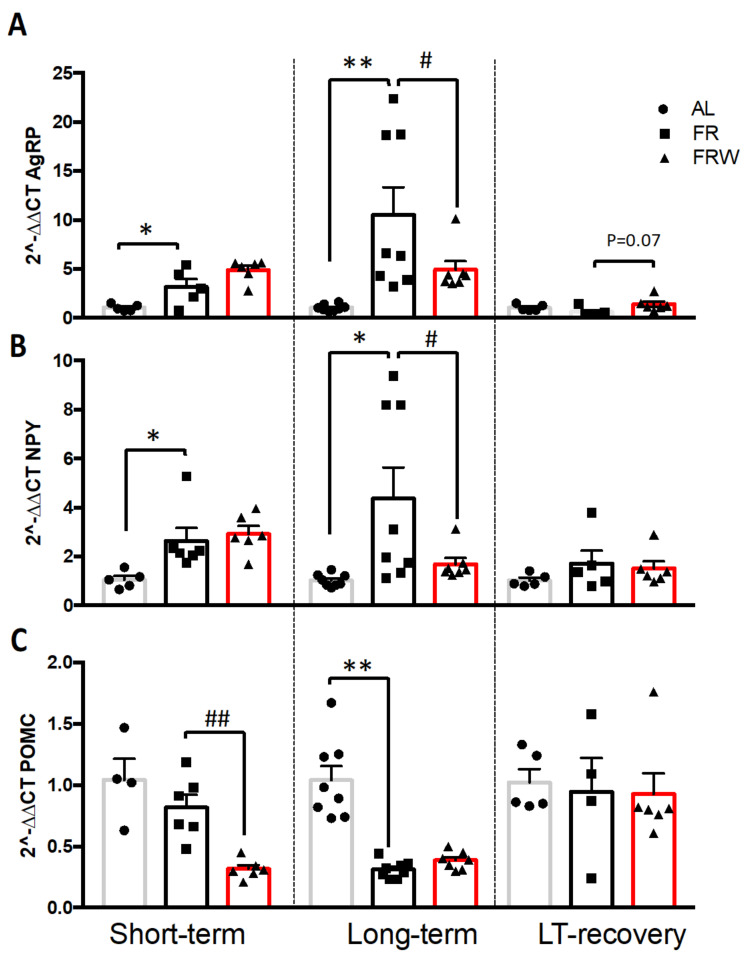
Impact of short-term, long-term (LT) and nutritional recovery (LT-Rec) on the mRNA expression of hypothalamic of agouti-related peptides (**A**, AgRP), neuropeptide Y (**B**, NPY) and pro-opiomelanocortin (**C**, POMC). AL: *ad libitum*; FR: food restricted; FRW: food restricted with wheel. Data represent mean ± SEM (*n* = 4–8 per group). * *p* < 0.05 AL vs. FR; # *p* < 0.05 FR vs. FRW; ** *p* < 0.01 AL vs. FR; ## *p* < 0.01 FR vs. FRW.3.2.5. mRNA expression receptors of leptin, ghrelin and AgRP/NPY and melanocortin systems in the hypothalamus.

**Figure 6 nutrients-13-02786-f006:**
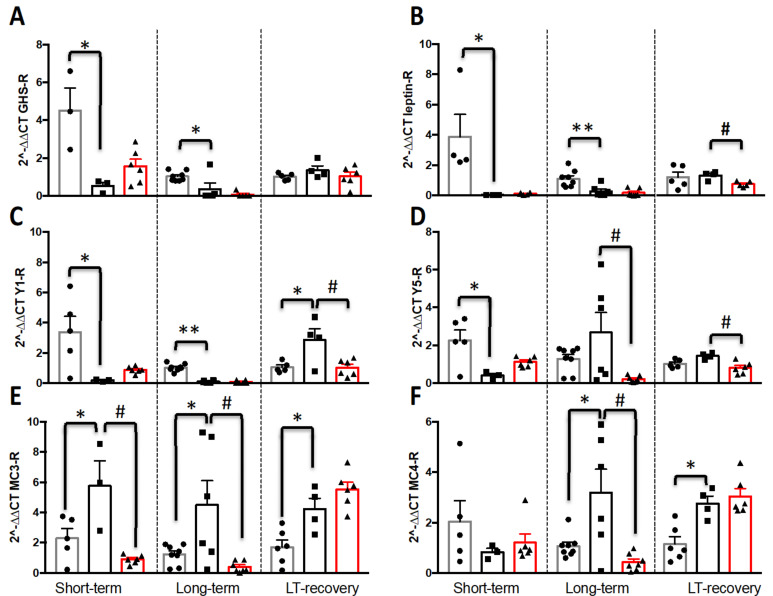
Impact of short-term, long-term (LT) and nutritional recovery (LT-Rec) on the mRNA expression of hypothalamic of (**A**) ghrelin receptor (GHSR), (**B**) leptin receptor, (**C**) Y1-receptor, (**D**) Y5-receptor, (**E**)melanocortin 3-receptor, (**F**) melanocortin 4-receptor. AL: *ad libitum*; FR: food restricted; FRW: food restricted with wheel. Data represent mean ± SEM (*n* = 3–8 per group). * *p* < 0.05 AL vs. FR; # *p* < 0.05 FR vs. FRW; ** *p* < 0.01 AL vs. FR.

**Figure 7 nutrients-13-02786-f007:**
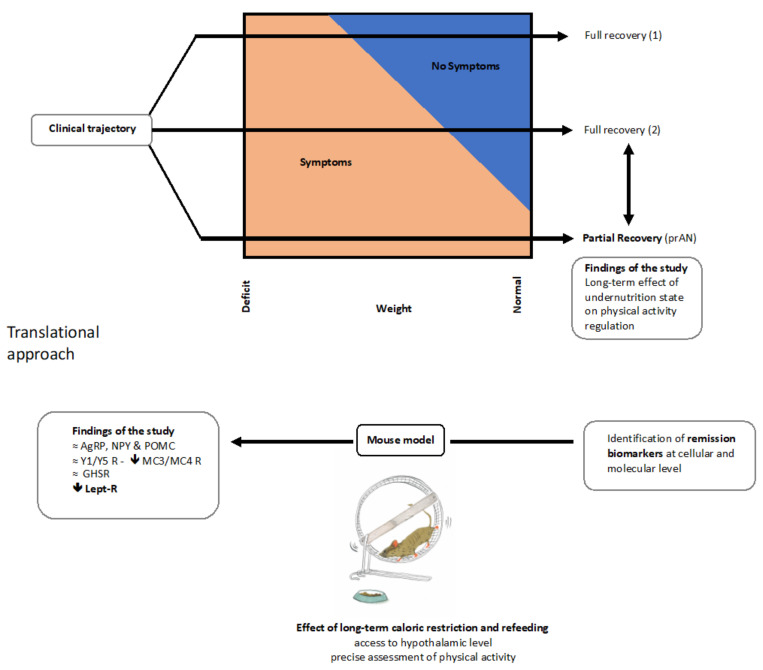
A translational approach of recovery in AN. Weight restoration is an important step of clinical management of patients suffering from AN. However, there is a high degree of inter-individual variability in the response to weight loss correction. Thus, some patients recover from their symptoms as soon as they start weight restoration (1) and others require significant weight gain to see their symptoms improving (2). Others still have symptoms despite full weight recovery: it is partial recovery (prAN). Our initial data on patients with prAN describe a long-term effect of undernutrition state on physical activity regulation. Beyond the recovery of a normal weight, it is necessary to describe objective biomarkers of remission. To identify them at the cellular or molecular levels, mouse models mimicking the effect of prolonged weight deficit on brain tissues after weight recovery is proving very useful. We demonstrated here the persistence of alteration of the central leptinergic pathway ten days after weight recovery. AgRP: agouti related-peptide; NPY: neuropeptide Y; POMC: pro-opiomelanocortin; Y1-R/Y5-R: neuropeptide Y receptor 1/receptor 5; MC3R/MC4-R: melanocortin 3 and 4 receptors; GHSR: ghrelin receptor; Lept-R: leptin receptor.

**Table 1 nutrients-13-02786-t001:** List of PCR primers. AgRP, agouti-related peptide; GAPDH, glyceraldehyde-3-phosphate dehydrogenase; GHSR, ghrelin receptor, Lept-R, leptin receptor; MC3-R/MC4-R, melanocortin 3 and 4 receptors; NPY, neuropeptide Y; POMC, proopiomelanocortin; Y1-R/Y5-R, neuropeptide Y receptor 1/receptor 5.

Gene	Reference	Forward Primer	Reverse Primer
AgRP	U89486.1	5′-CCCAGAGTTCCCAGGTCTAAGTCT-3′	5′-CACCTCCGCCAAAGCTTCT-3′
NPY	NM_023456.2	5′-CCGCTCTGCGACACTACAT-3′	5′-TGTCTCAGGGCTGGATCTCT-3′
POMC	BC061215.1	5′-AGTGCCAGGACCTCACCA-3′	5′-CAGCGAGAGGTCGAGTTTG-3′
MC3R	NM_008561	5′-TCCGATGCTGCCTAACCTCT-3′	5′-GGATGTTTTCCATCAGACTGACG-3′
MC4R	NM_016977	5′-CCCGGACGGAGGATGCTAT-3′	5′-TCGCCACGATCACTAGAATGT-3′
Y1R	NM_010934	5′-TGATCTCCACCTGCGTCAAC-3′	5′-ATGGCTATGGTCTCGTAGTCAT-3′
Y2R	NM_008731	5′-GCCAGGGCACACTACTCCTA-3′	5′-CTACCCCTAGCAAGATGATGGA-3′
Y5R	NM_016708.3	5′-CATCTCAAGCAGAAGCGACC-3′	5′-CTCCATACTAGAGTCCTCGGG-3′
Lept-R	NM_010704.2	5′-GTCTTCGGGGATGTGAATGTC-3′	5′-ACCTAAGGGTGGATCGGGTTT-3′
GHSR	NM_177330	5′-TGGAGATCGCGCAGATCAG-3′	5′CCGGGAACTCTCATCCTTCAG-3′
GADPH	GU214026	5′-GAACATCATCCCTGCATCC-3′	5′-CCAGTGAGCTTCCCGTTCA-3′

**Table 2 nutrients-13-02786-t002:** Clinical characteristics of Anorexia Nervosa patients (AN) and partially recovered AN (prAN). Significant *p*-values are in bold.

	AN (*n* = 225)		prAN (*n* = 41)		AN vs. prAN	
	Mean	SD	Mean	SD	U	*p*
Age	27.8	9.76	27.2	8.43	4558	0.904
BMI	15.1	1.19	20.5	1.81	0	**<0.001**
BMI min	13.4	3.07	15	1.34	1956	**<0.001**
Illness duration	8.9	8.8	10.3	8.6	3274	0.199
EDI						
Drive for thinness	10.5	6.95	13.8	6.33	3030	**0.004**
Bulimia	3.65	5.35	5.92	6.37	3290	**0.017**
Body dissatisfaction	12.1	7.16	16.9	8.05	2750	**<0.001**
Ineffectiveness	11.6	7.57	12.7	6.52	3890	0.383
Perfectionism	7.52	4.76	9.15	4.51	3407	**0.044**
Interpersonal distrust	6.29	4.59	6	4.56	4016	0.564
Interoceptive awareness	10.5	6.9	13.7	7.05	3143	**0.008**
Maturity fears	6.87	5.95	7.35	6.22	4040	0.604
Ascetism	7.73	4.95	8.68	4.49	3666	0.161
Emotional dysregulation	5.44	5.49	8.25	6.72	3185	**0.011**
Social insecurity	7.84	5.23	8.4	4.26	3904	0.402
Total	89.62	46.57	110.97	41.14	3133	**0.007**
HADS Anxiety	12.5	4.33	13.9	3.45	2969	0.069
HADS Depression	8.59	4.53	8.29	4.34	3485	0.663
YBS Compulsion	7.84	5.37	8.74	4.27	3410	0.525
YBS Obsession	8.93	5.01	10.6	4.16	3061	0.117
Ferritin (ng/mL)	81	75.1	30.8	17	845	**<0.001**
Albumin (g/L)	47.7	6.21	43.7	8.73	1670	**<0.001**

AN: Anorexia Nervosa; BMI: body mass index; EDI: Eating Disorder Inventory II; HADS: Hospital Anxiety and Depression scale; prAN: partially recovered Anorexia Nervosa; YBS: Yale Brown Scale.

**Table 3 nutrients-13-02786-t003:** Representativity of the sample with cardiac monitoring. Comparison of clinical characteristics of all patients with anorexia nervosa (AN), acute AN (aAN) or partially recovered AN (prAN) and the sample with continuous cardiac monitoring. Significant *p*-values are in bold.

	AN (*n* = 266) vs. AN with Cardiac Monitoring (*n* = 68)		aAN (*n* = 225) vs. aAN with Cardiac Monitoring (*n* = 55)		prAN (*n* = 41) vs. prAN with Cardiac Monitoring (*n* = 13)	
	U	*p*	U	*p*	U	*P*
Age	7656	0.108	5030	0.050	218	0.559
BMI	7786	0.110	5256	0.124	209	0.248
BMI min	6640	0.371	4258	0.338	265	0.984
Illness duration	6721	**0.032**	4517	**0.034**	220	0.759
EDI						
Drive for thinness	7989	0.908	5456	0.910	224	0.727
Bulimia	7936	0.889	5408	0.825	232	0.860
Body dissatisfaction	7483	0.373	4968	0.270	238	0.965
Ineffectiveness	7927	0.834	5408	0.834	237	0.957
Perfectionism	7698	0.574	5149	0.462	215	0.586
Interpersonal distrust	6987	0.098	4841	0.172	195	0.326
Interoceptive Awareness	7784	0.668	5447	0.896	157	0.071
Maturity fears	7664	0.539	5170	0.488	240	1.000
Ascetism	7816	0.704	5500	0.981	191	0.290
Emotional Dysregulation	7170	0.169	5120	0.425	176	0.167
Social insecurity	7430	0.331	4986	0.285	232	0.861
Total	7552	0.460	5200	0.560	211	0.529
HADS Anxiety	6356	0.104	4672	0.512	125	**0.019**
HADS Depression	7043	0.634	4890	0.865	192	0.418
YBS Compulsion	7126	0.697	4948	0.922	200	0.531
YBS Obsession	6751	0.310	4610	0.397	207	0.629

aAN: acute Anorexia Nervosa; AN: Anorexia Nervosa; BMI: body mass index; EDI: Eating Disorder Inventory II; HADS: Hospital Anxiety and Depression scale; prAN: partially recovered Anorexia Nervosa; YBS: Yale Brown Scale.

**Table 4 nutrients-13-02786-t004:** Non parametric correlation for BMI minimum, current BMI, illness duration and heart rate ranges in all patients with (AN *n* = 68). Significant *p*-values are in bold. BMI: body mass index; EDI: Eating Disorder Inventory II; HADS: Hospital Anxiety and Depression scale; YBS: Yale Brown Scale.

		BMI min	Current BMI	Illness Duration	Bradycardia	Physiological	Moderate	High Intensity
BMI min	Rho	—						
	*p*-value	—						
Current BMI	Rho	0.538	—					
	*p*-value	**<0.001**	—					
Illness duration	Rho	−0.131	−0.040	—				
	*p*-value	0.293	0.746	—				
Bradycardia	Rho	0.079	−0.143	−0.298	—			
	*p*-value	0.531	0.251	**0.015**	—			
Physiological	Rho	0.168	0.322	0.158	−0.642	—		
	*p*-value	0.181	**0.008**	0.206	**<0.001**	—		
Moderate	Rho	−0.234	−0.032	0.277	−0.733	0.343	—	
	*p*-value	0.061	0.800	**0.024**	**<0.001**	**0.005**	—	
High intensity	Rho	−0.271	*−*0.084	0.213	−0.557	−0.032	0.669	—
	*p*-value	**0.029**	0.502	0.087	**<0.001**	0.797	**<0.001**	—

**Table 5 nutrients-13-02786-t005:** Non parametric correlation for BMI minimum, current BMI, illness duration and heart rate ranges in acute AN patients (*n* = 55). Significant *p*-values are in bold. Trend values are in italics. BMI: body mass index; EDI: Eating Disorder Inventory II; HADS: Hospital Anxiety and Depression scale; YBS: Yale Brown Scale.

		BMI min	Current BMI	Illness Duration	Brady-Cardia	Physiological	Moderate	High Intensity
BMI min	Rho	—						
	*p*-value	—						
Current BMI	Rho	0.620	—					
	*p*-value	<**0.001**	—					
Illness duration	Rho	−0.186	−0.073	—				
	*p*-value	0.182	0.600	—				
Bradycardia	Rho	0.121	0.081	−0.423	—			
	*p*-value	0.394	0.566	**0.002**	—			
Physiological	Rho	0.050	0.173	0.195	−0.670	—		
	*p*-value	0.725	0.217	0.161	**<0.001**	—		
Moderate	Rho	−0.279	−0.188	0.324	−0.730	0.342	—	
	*p*-value	**0.045**	0.177	**0.018**	**<0.001**	**0.013**	—	
High intensity	Rho	−0.332	*−0.269*	0.307	−0.535	−0.059	0.677	—
	*p*-value	**0.016**	*0.052*	**0.025**	**<0.001**	0.675	**<0.001**	—

**Table 6 nutrients-13-02786-t006:** Non parametric correlation for BMI min, current BMI, illness duration and heart rate ranges in partially recovered AN patients (*n* = 13). Trend values are in italics. BMI: body mass index; EDI: Eating Disorder Inventory II; HADS: Hospital Anxiety and Depression scale; YBS: Yale Brown Scale.

		BMI min	Current BMI	Illness Duration	Brady-cardia	Physiological	Moderate	High Intensity
BMI min	Rho	—						
	*p*-value	—						
Current BMI	Rho	−0.234	—					
	*p*-value	0.442	—					
Illness duration	Rho	0.306	−0.113	—				
	*p*-value	0.310	0.714	—				
Bradycardia	Rho	0.479	−0.126	−0.415	—			
	*p*-value	0.098	0.683	0.158	—			
Physiological	Rho	0.193	0.297	−0.116	−0.462	—		
	*p*-value	0.528	0.325	0.707	0.115	—		
Moderate	Rho	−0.303	−0.121	0.072	−0.363	0.044	—	
	*p*-value	0.315	0.696	0.816	0.224	0.892	—	
High intensity	Rho	*−0.525*	−0.099	−0.487	*−0.549*	−0.198	0.275	—
	*p*-value	*0.065*	0.751	0.091	*0.055*	0.517	0.363	—

**Table 7 nutrients-13-02786-t007:** Acyl ghrelin, desacyl ghrelin and leptin plasma levels in short-term, long-term and recovery mice protocols. Significant values are in bold. Data are expressed in mean +/− SEM or *p*-value (*p*). AL: *ad libitum*; FR: food restricted; FRW: food restricted and wheel. AL versus FR: * *p* < 0.05; ** *p* < 0.01.

	Acyl-Ghrelin	Desacyl-Ghrelin	Leptin
**Short term**			
AL	229+/−35.4	679+/−79.8	5724+/−866
FR	576+/−63.7 **	1583+/−160.5 *	3548+/−248.9 *
FRW	958+/−165.7	1763+/−252.7	3080+/−202.7
*Group effect (p)*	**<0.001**	**<0.001**	**0.007**
**Long term**			
AL	166+/−42.5	1085+/−132.4	4668+/−419.8
FR	147+/−20.3	1459+/−108.4	3365+/−155.5 **
FRW	240+/−41.5	1534+/−132.2	3518+/−88.3
*Group effect (p)*	0.208	**0.045**	**0.004**
**Long term-Recovery**			
AL	150+/−14.6	790+/−13.7	4594+/−380.7
FR	137+/−27.3	913+/−125.7	3639+/−95.4 *
FRW	115+/−14	728+/−40.7	2915+/−100.9
*Group effect (p)*	0.633	0.433	**<0.001**

## Data Availability

The data that support the findings of this study are not available.

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
