# Peer review of "Exploring the Mechanisms of Recovery in Anorexia Nervosa through a Translational Approach: From Original Ecological Measurements in Human to Brain Tissue Analyses in Mice"

_nutrients, 2021, doi:10.3390/nu13082786_

Round 1
Reviewer 1 Report
Duriez et al. present an interesting analysis of both clinical and preclinical data looking at behavioral and physiological markers of recovery from anorexia nervosa. I commend the authors on taking on translational project, implanting both mice and humans.
While I believe this article adds important knowledge to the understanding of eating disorders, there are a few things the authors should address, as outlined here.
Major comments:
- The authors went through a great deal of work to include both mice and humans, but the way the paper is written very much segregates the findings, almost like reading two unrelated manuscripts that have been pushed together. Is there a way to better integrate the rational, data and discussion to make the paper read as a cohesive study? What does the animal data say about the human condition and vice versa?
- It seems that an important control group, mice with ad libitum food access and running wheel, is missing. Have the authors considered adding this group? This should (minimally) be added as a restriction in the discussion of the study.
- There is some “clean-up” work needed in the presented data. All data given as percent should have error values. Graphs should be more clearly labeled (ex. Figure 2A the left graph has no x-axis label, the right graph is labeled “days after nutritional recovery”—but then gives values such as 66, 67,68 etc). The x-scale on graph 2E is misleading, because it appears incremental, but is not. Section 3.2.2 seems to be lacking full statistical reporting. Finally on Figure 3, it’s unclear how a 18g mouse can have ~250g of fat.
- I would encourage the authors to add more details to the preclinical “methods” section. While they state that data was removed from analysis if the authors felt the cage data did not reflect the data of individual mice. What were the criteria for this determination? How many animals/cages were removed from analysis. How many mice were there to start? Do the reported N values include these animals? While it says no mice died—were any mice removed for low bodyweight? How often was bodyweight measured?
Minor comments:
- Please provide more information on the clinical population (ex. sex, age etc). Was it all females, and is that what motivated an all-female animal group?
- Because this article may attract clinical and basic science readers, it might be a good idea to add a little more detail to the text so that all individuals can understand the measures. For example, on page 7, line 281, anxiety scores are reported, but it’s unclear if higher or lower scores indicate more anxiety.
- On page 12, lines 340, it is unclear how percent recovery is calculated—can you spell this out like on figure 2C.
- Graph 2F is interesting, but I wonder if there’s a better way to graph the data that will simultaneously capture the fact that the diurnal rhythm is becoming more regular in recovery, but simultaneously the overall running is decreasing.
- In figure 3, why do the AL mice from short term have more fat than the AL mice in the long term and LT recovery groups? This doesn’t seem to make much sense to me?
- Because it’s hard to make out the images in figure 4, can some sort of quantification be added?
- There’s a great deal of variability in some of the graphs in figures 5 and 6, it would be interesting to add the individual data points to these graphs so the reader can interpret where that variability is coming from.
Author Response
We first thank the reviewer for his/her careful reading and the helpful and relevant comments made on our manuscript. We have answered the questions and suggestions in the following paragraphs (see the file join to this message). All changes are highlighted in green in the manuscript.

Reviewer 2 Report
This is an excellent manuscript reporting research in humans and animals. The presented data may help to disentangle the relations between body weight, body weight regulating hormones and physical activity. This has clinical implications.
The manuscript is well written. The study design is appropriately explained. The data support the conclusions. The figures and tables help the reader to understand the procedures. Limitations are mentioned.
I have only minor comments regarding the misspelling of some words:
Page 3, line 127 and table 2:
“ferritine” would correctly be spelled “ferritin”.
Page 14, line 356:
“nutritonal recovery”: There is an “i” missing in “nutritional”.
Page 15, legend of figure 3:
“Subcutaneaous fat mass and …” there is a misspelling in “Subcutaneous” with one “a” too much.
Page 19, line 510:
“… ghrelin and AgRP/NPY and melacortin”: There is an “n” missing in “melanocortin”.
Page 21, line 606:
“It could be considered like a sequaella of undernutrition state” should read “… sequelae …”
Page 21, lines 621 and 622:
“… weight gain during nutritional recovery. Indeed, . Indeed, using a rat model of AN (ABA model), Giles et al. (2016) demonstrated”: One “Indeed” should be deleted.
Page 22, line 657:
“… their mRNA expression were strongly”: Should be singular “their mRNA expression was strongly…”
Author Response
We thank the reviewer for his positive comments and his very careful reading. We have taken into account all minor comments. All changes are highlighted in yellow in the manuscript.

Round 2
Reviewer 1 Report
Thank you for the detailed responses to each concern.